# Organocatalyst-mediated five-pot synthesis of (−)-quinine

Takahiro Terunuma [1] & Yujiro Hayashi [1]✉

In this work, the enantioselective total synthesis of (−)-quinine has been accomplished in a pot-economical manner using five reaction vessels. In the first pot, reactions involve the diphenylprolinol silyl ether-mediated Michael reaction, aza-Henry reaction, hemiaminalization, and elimination of $HNO_2$ (five reactions), affording a chiral tetrahydropyridine with excellent enantioselectivity. In the second pot, five reactions proceed with excellent diastereoselectivity to afford a trisubstituted piperidine with the desired stereochemistry. A further five reactions are carried out in the last one-pot sequence.

Quinine is one of the cinchona alkaloids; it is extracted from the bark of cinchona trees[1]. It is an important antimalarial medication[2]. Its derivatives are used not only as ligands of asymmetric metal catalysts, but also as effective asymmetric organocatalysts[3–10]. Quinine has fascinated synthetic organic chemists for several decades[11–28]. It contains four chiral centers, and stereoselective synthesis is a challenge.

In 1944, Woodward and Doering synthesized homomeroquinene—the first formal synthesis of a racemic quinine[4,5]. Stork then reported the first synthesis of chiral (−)-quinine from a chiral starting material using highly diastereoselective reactions[17]. Jacobsen[18] and Kobayashi[20] independently synthesized (−)-quinine by constructing a quinuclidine skeleton, by a C8−N bond formation, using an epoxide as an electrophile. Jacobsen used an asymmetric catalytic reaction for the enantioselective synthesis. Aggarwal used a chiral Corey−Chaykovsky reaction as a key step[22,23], while Maulide used a C−H activation reaction and an aldol reaction[25]. Chen synthesized quinine and quinidine using a local desymmetrization-based strategy[26].

Recently, organocatalysts have been used for the synthesis of quinine. Hatakeyama used a proline-mediated intramolecular aldol reaction for the formation of a piperidine skeleton[24], while the Michael reaction of dimethyl malonate and α,β-unsaturated aldehyde catalyzed by diphenylprolinol silyl ether is a key step in Córdova's synthesis[27]. Ishikawa has reported a synthesis of (−)-quinine involving a one-pot synthesis of a chiral piperidine skeleton by a diphenylprolinol silyl ether-mediated formal aza-[3 + 3] cycloaddition/Strecker-type cyanation reaction[28]. Although there are several asymmetric total syntheses of (−)-quinine, it is still desirable to develop an efficient and practical method for its synthesis. Moreover, a method that facilitates derivatization is desirable, particularly because of its importance as a medication and as an organocatalyst.

One-pot operations are effective methods for making several bonds and generating a complexity of molecules in a single-pot sequence[29–31]. Moreover, one-pot operations circumvent several purification steps via in situ quenching events, thereby minimizing chemical waste generation and saving time. Based on this, we proposed the concept of "pot economy." We subsequently accomplished several total syntheses based on this concept[32–38].

In this work, we describe the "pot-economical" enantioselective total synthesis of (−)-quinine using organocatalyst-mediated reaction.

## Results

### The retrosynthetic analysis of (−)-quinine

The retrosynthetic analysis of (−)-quinine is shown in Fig. 1. It would be synthesized by making a C6−N bond from a piperidine precursor 2 via an intramolecular $S_N2$ reaction. 2 would be prepared by the addition reaction of a quinoline unit to piperdinecarbaldehyde 3. 3 would be synthesized from a substituted piperidine derivative 4 by functional group transformation. 4 would be prepared by a three-component coupling reaction of aldehyde 5, nitroalkene 6, and imine 7, as developed by our group[39].

Earlier, we reported a one-pot synthesis of a piperidine skeleton via a sequential reaction of the diphenylprolinol silyl ether-mediated Michael reaction[40], aza-Henry reaction, and hemiaminalization reaction (Fig. 2). As simple starting materials were used in the earlier study, we considered it now a challenge to determine whether the key

[1]Department of Chemistry, Graduate School of Science, Tohoku University, 6-3 Aramaki Aza-Aoba, Aoba-ku, Sendai, Miyagi 980-8578, Japan.
✉e-mail: yujiro.hayashi.b7@tohoku.ac.jp

**Fig. 1 | Retrosynthetic analysis of (−)-quinine.** Organocatalyst-mediated pot-economical synthesis of (−)-quinine (**1**). TBDPS *t*-butyldiphenylsilyl.

**Fig. 2 | Previous work.** One-pot synthesis of a piperidine skeleton via a sequential reaction of the diphenylprolinol silyl ether-mediated Michael reaction, aza-Henry reaction, and hemiaminalization reaction. Et ethyl, Ns *p*-nitrophenylsulfonyl, Ph phenyl, TFA trifluoroacetic acid, TMS trimethylsilyl.

**Fig. 3 | Diphenylprolinol sily ether-mediated asymmetric Michael reaction.** Asymmetric Michael reaction of **5** and **6** using diphenylprolinol silyl ether. Enantiomeric excess (ee) was determined after Wittig reaction. *'*Bu *t*-butyl.

reaction proceeds using densely functionalized substrates and whether high stereoselectivity is realized.

## The first pot reaction

4-Arylthiobutanal **5**[41] was selected as a starting aldehyde, taking into consideration of the required introduction of a C10–C11 double bond in a latter stage of the synthesis. The asymmetric Michael reaction of **5** and nitroalkene **6**, catalyzed by diphenylprolinol silyl ether, proceeded to afford **8** in good yield and with excellent diastereo- and enantioselectivities (Fig. 3). However, the next aza-Henry reaction was found to be problematic. In our previous study, we used *p*-nitrophenylsulfonyl (Ns)[42] imine generated from arylaldehyde (Fig. 2). The Ns imine of ethyl glyoxylate **7a** is too unstable and difficult to handle (Fig. 4). Imine **7b** derived from *p*-anisidine is too stable to react with **8**. On the other

hand, the corresponding *N*-Boc imine **7c** possesses suitable stability and reactivity, but it is difficult to isolate it. Thus, it was generated in situ from the corresponding sulfonyl precursor **9**[43] with DBU. A domino reaction of **8** and **9**, involving the generation of the imine and the aza-Henry reaction, proceeded well to afford piperidine **4** and tetrahydropyridine **10** in 35% and 50% yield, respectively (Fig. 5). Although **4** was an expected product, **10** would be a more suitable intermediate for the total synthesis, because of the harsh reaction conditions that are usually necessary for the reductive removal of a NO₂ group from **4**[44–46]. **4** can be converted to **10** by the treatment with DBU.

These reactions (**5** + **6**→**10**) can be conducted in a single pot (Fig. 6). The first pot reaction starts from the Michael reaction. The asymmetric Michael reaction of **5** and **6** proceeded to afford product **8**.

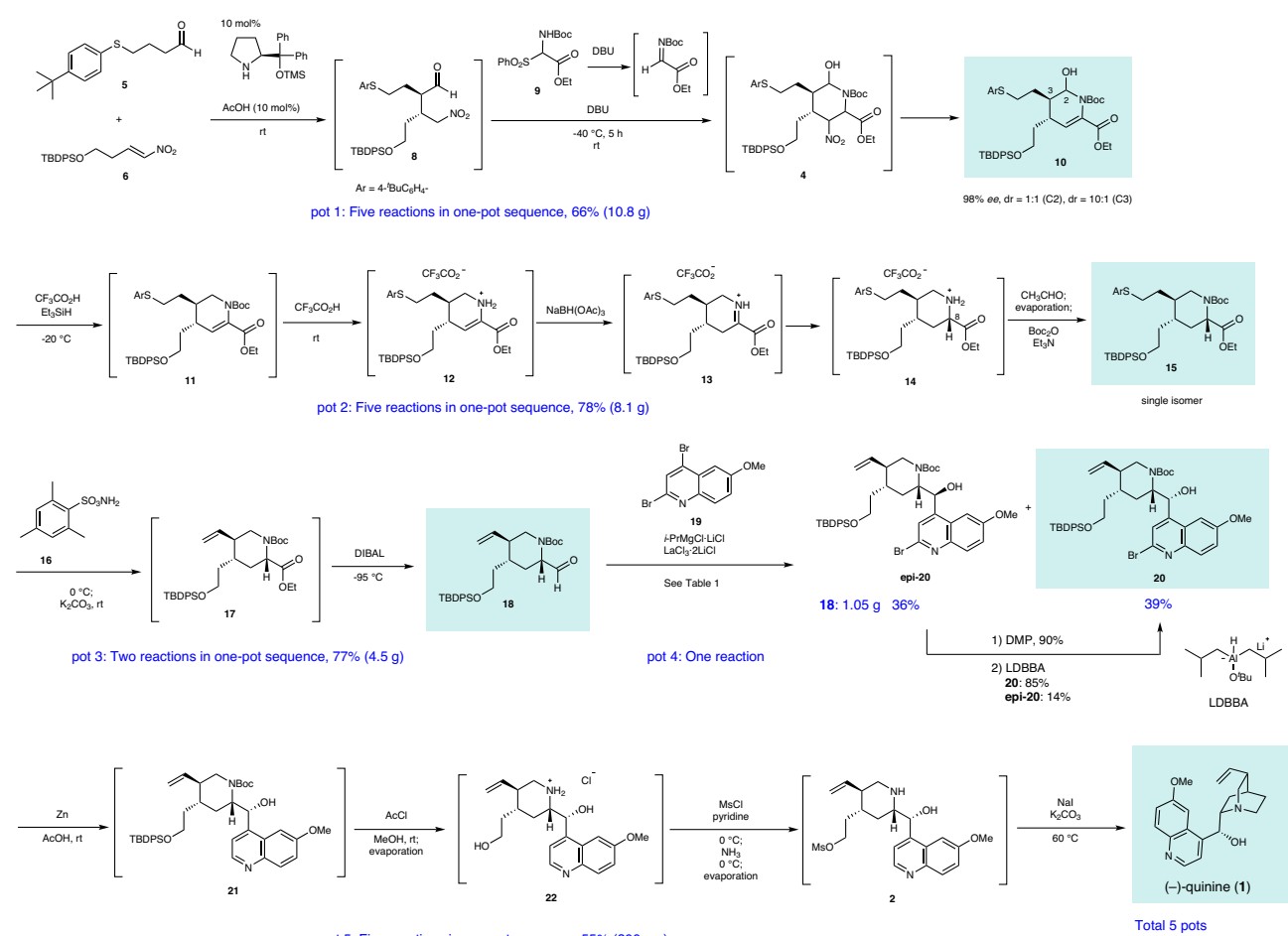

**Fig. 4 | Imines and the precursor of imine.** In this work, **7a-7c**, and **9** were investigated. Boc *t*-butoxycarbonyl, Me methyl.

**Fig. 5 | Aza-Henry, hemiaminalization, and elimination reaction.** In situ generation of imine from **9** using DBU was the key to obtain **4** and **10**. **4** could be converted to **10** by DBU. DBU 1,8-diazabicyclo[5.4.0]undec-7-ene.

**Fig. 6 | Five-pot asymmetric synthesis of (−)-quinine.** Organocatalyst-mediated construction of the piperidine ring in one-pot. After two one-pot reaction and introduction of quinoline ring, sequential deprotection, mesylation and cyclization afforded (−)-quinine (**1**). Ac acetyl, DIBAL diisobutylaluminium hydride, LDBBA lithium diisobutyl-*tert*-butoxyaluminum hydride, Ms methanesulfonyl, Pr propyl.

Upon the addition of imine precursor **9** and DBU to the same vessel, a domino reaction proceeded. That is, in situ imine generation occurred, and an aza-Henry reaction proceeded, sequentially, followed by hemiaminalization to afford **4**. Elimination of $HNO_2$ with DBU afforded **10** with a mixture of diastereomers at C2. Good overall yield, high diastereoselectivity (dr = 10:1 at C3), and excellent enantioselectivity (98% *ee*) were obtained. The reaction proceeded deca-gram scale efficiently. It should be noted that DBU plays three roles as a base in this transformation: (1) to generate imine (**9**→**7c**), (2) to promote the aza-Henry reaction, and (3) to eliminate $HNO_2$.

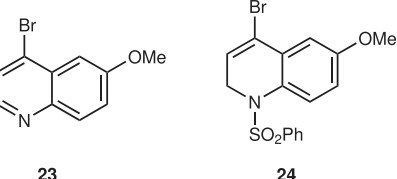

**Fig. 7 | Bromoquinoline derivatives.** Bromoquinoline derivatives were investigated by Ishikawa's group.

## The second pot reaction

The second pot reaction starts with the reductive removal of a hydroxy group at C2 by the treatment of **10** with $Et_3SiH$ and $CF_3CO_2H$ (TFA) at −20 °C[39]. Upon increasing the temperature of a reaction mixture to room temperature, the deprotection of a Boc group took place, to afford an enamine **12**, which was tautomerized to afford an iminium ion **13**. Upon the addition of $NaBH(OAc)_3$ to the reaction vessel, stereoselective reduction occurred to afford **14**[17]. Addition of $CH_3CHO$ led to decomposition of the remaining reducing reagent, and the excess $CH_3CHO$ and TFA were removed under reduced pressure. Thus, through this in situ quenching method[30], a one-pot sequential reaction can be realized.

Further addition of $Boc_2O$ and $Et_3N$ to the reaction mixture afforded a highly substituted piperidine derivative **15** as a single isomer with the desired configuration. It should be noted that TFA plays three roles as an acid. It facilitates the formation of tetrahydropyridium from hemiaminal **10** at low temperature (−20 °C), to deprotect the Boc group at higher temperature, and to generate the iminium ion **13**. It is synthetically efficient that the same reagent plays several roles in a one-pot reaction[29]. The reaction proceeded in a gram scale and the overall yield of this second pot (**10**→**15**) was 78%. In this transformation, procedures of deprotection and reprotection of Boc group were involved. The stereoselective reduction of Boc protected alkene **11** did not proceed well, while stereoselective reduction of the iminium ion **13** was successful. Thus, the deprotection step of Boc group is necessary.

We employed two reducing reagents such as $Et_3SiH$ and $NaBH(OAc)_3$ for the reduction of hemiaminal (**10**→**11**) and iminium ion (**13**→**14**). It would be synthetically simple to use a single reducing reagent in a one-pot reaction. Although $Et_3SiH$ can act as a reducing reagent in both reactions, the second reaction was slow in the low yield with the partial deprotection of TBDPS group. In the case of $NaBH(OAc)_3$ the reaction from **10** to **11** did not proceed well with the generation of several byproducts. On the other hand, the sequential use of two reducing reagents gave a good result in the second pot reaction.

## The third pot reaction

The third pot involves the preparation of a C10−C11 alkene and half-reduction of an ester. When arylthiol ether **15** was treated with **16**, according to Matsuo's protocol, alkene **17** was generated[47]. In the same reaction vessel, the addition of DIBAL at low temperature (−95 °C) reduced an ester, and aldehyde **18** was obtained in good yield (77%) over two steps, in one pot. The conventional method of arylsulfanyl-ethyl moiety into vinyl substituent is a two-step process: oxidation to sulfoxide and elimination of ArSOH. However, by applying Matsuo's protocol, it becomes a one-step process. Moreover, the reaction conditions here are comparable to those of the one-pot sequential reaction of the next DIBAL reduction. This step is also scalable.

## The fourth pot reaction

The next step is the introduction of a quinoline moiety to aldehyde **18**—this was found to be very difficult. According to Ishikawa, several quinoline metals generated from **23** did not react with a similar aldehyde with formation of dimer of **23** because the C2 position of quinoline is electrophilic—thus, dihydroquinoline lithium reagent

generated from **24** was developed as a nucleophile (Fig. 7)[28]. In order to synthesize C2′ derivatives of quinine and also to reduce the electrophilicity at the C2 of quinoline, we used 2,4-dibromo-6-methoxyquinoline (**19**) as a quinoline precursor. However, the selective halogen−metal exchange reaction of Br at C4 over Br at C2 is another concern. Pleasingly, the selective halogen−metal exchange of Br at C4 occurred selectively. But neither lithium nor magnesium species were added to aldehyde **18** (Table 1; entries 1, 2).

Then, the use of an additive was investigated. Whereas $ZnCl_2$[48] did not afford the desired product (entry 3), a moderate yield was obtained in the presence of $CeCl_3$[49,50] (entry 4). After numerous investigations, we achieved success with $LaCl_3·2LiCl$[51]; the addition products were obtained in good yield (**20**: 41%, **epi-20**: 44%, entry 5). BuLi could be used for the first halogen−metal exchange reaction, but the yield was low (entry 7). The reaction also proceeded in a gram scale (entry 6). Although the undesired diastereomer **epi-20** was obtained, **epi-20** can be converted to the desired isomer **20** via oxidation with DMP (90%) and stereoselective reduction using LDBBA (lithium diisobutyl-*tert*-butoxyaluminum hydride)[52,53], in which **20** was obtained in 85% with **epi-20** (14%).

## The fifth pot reaction

The last pot reaction begins with **20**. Upon treatment of **20** with Zn and acetic acid, the C−Br bond was reduced to a C−H bond to afford **21**[23]. Addition of AcCl and MeOH to the reaction mixture generated HCl, which then deprotected both the Boc group and the TBDPS group, to afford piperidine **22**. Using MsCl and pyridine as a base, selective mesylation of a primary alcohol in the presence of a secondary alcohol and piperidine took place[17], in which piperidine was protected as its HCl salt. Upon the addition of ammonia, the piperidine·HCl salt was converted to free piperidine **2**, and excess ammonia was removed under reduced pressure. An intramolecular $S_N2$ reaction proceeded in the presence of NaI and $K_2CO_3$ at 60 °C to afford (−)-quinine (**1**). The physical properties of the synthetic (−)-quinine (**1**) were identical in all respects to the reported data[18]. The yield of the last pot reaction (**20**→**1**) was 55%.

The order of the reaction steps and reaction conditions are essential for the success of the last one-pot operations. There are several other reaction conditions suitable for each reaction step; however, a combination of these steps in a single reaction vessel presents difficulties. For instance, Zn reduction of bromoquinoline did not proceed well after the formation of a quinuclidine skeleton. Reduction of bromoquinoline **20** proceeded well using $NaBH_4$ and TMEDA in the presence of a catalytic amount of $PdCl_2(dppf)$[54], but subsequent deprotection of TBDPS and Boc did not proceed in a one-pot operation. The TBDPS group could be selectively deprotected without affecting the Boc group by the treatment of *n*-$Bu_4NF$, but the next mesylation could not proceed in the presence of *n*-$Bu_4NF$.

## The derivatization of (−)-quinine

As Br(C2)-quinoline derivative **19** was employed as a nucleophile, derivatization of (−)-quinine at the C2′ position would be possible. Br(C2′)-(−)-quinine **25**[55] was synthesized from **20** by a method similar to that of our last pot reaction for the synthesis of (−)-quinine, except for the Zn reduction. Four reaction steps, i.e., the deprotection of both

**Table 1 | Addition reaction of a quinoline derivative to aldehyde 18[a]**

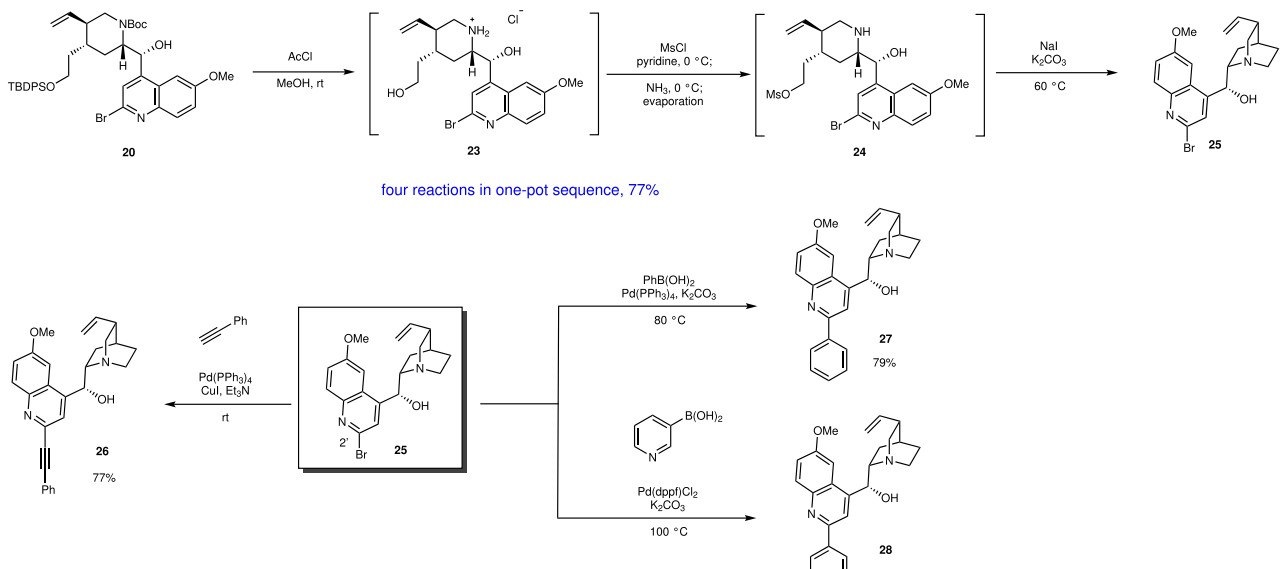

| Entry | Condition | Temp. [°C] | Yield [%][b] | |
|---|---|---|---|---|
| | | | **20** | **epi-20** |
| 1 | n-BuLi | −78 | <5 | <5 |
| 2 | i-PrMgCl·LiCl | −20 | <5 | <5 |
| 3 | i-PrMgCl·LiCl, ZnCl₂[c] | −20 | <5 | <5 |
| 4 | i-PrMgCl·LiCl, CeCl₃[d] | −20 | 27 | 26 |
| 5 | i-PrMgCl·LiCl, LaCl₃·2LiCl[e] | −20 | 41 | 44 |
| 6[f] | i-PrMgCl·LiCl, LaCl₃·2LiCl[e] | −20 | 39 | 36 |
| 7 | n-BuLi, LaCl₃·2LiCl | −78 | 32 | 33 |

[a]Otherwise noted, the reactions were performed using aldehyde **18** (0.12 mmol), dibromoquinoline **19** (0.36 mmol), n-BuLi (0.36 mmol) or i-PrMgCl·LiCl (0.34 mmol) in THF (0.1 M) at indicated temperature under $N_2$. Reaction was quenched with saturated $NH_4Cl$ solution.
[b]Isolated yield.
[c]ZnCl₂ (0.36 mmol) was used.
[d]CeCl₃ (0.36 mmol) was used.
[e]LaCl₃·2LiCl (0.36 mmol) was used.
[f]**18** (1.05 g) was used.

**Fig. 8 | Derivatization of (−)-quinine.** 25 was synthesized from **20** in one-pot, and it could be converted to quinine derivatives **26**–**28**.

(1) TBDPS and (2) Boc groups, (3) mesylation of a primary alcohol, and (4) an intramolecular $S_N2$ reaction, proceeded efficiently to afford **25** in good overall yield (77%) in a single vessel (Fig. 8). The introduction of alkyne, aryl, and heteroaryl moieties was successful, affording **26**, **27**, and **28**, respectively in good yields, by the Sonogashira reaction and Suzuki reaction.

## Discussion

In summary, an efficient, enantioselective, and pot-economical total synthesis of (−)-quinine has been accomplished. The present synthesis has several noteworthy features. (1) A highly functionalized chiral tetrahydropyridine can be synthesized in the first one-pot operation, which consists of a succession of five reaction steps, including a diphenylprolinol silyl ether-mediated asymmetric Michael reaction as developed in our group, a domino aza-Henry reaction/hemiacetal formation involving the in situ generation of imine, and elimination of $HNO_2$. (2) Five reactions can be successfully conducted in the second one-pot operation: the reductive removal of a hydroxy group, deprotection of Boc, iminium ion generation, stereoselective reduction, and Boc protection, thus affording the trisubstituted piperidine as a single isomer with the correct configuration. (3) Five reactions can proceed efficiently by careful choice of the reaction order and the reaction

conditions: the reductive removal of Br, deprotection of Boc and TBDPS groups, selective synthesis of a mono-Ms derivative, and an intramolecular $S_N2$ reaction. (4) The use of 2,4-dibromo-6-methoxy quinoline for the introduction of a quinoline moiety makes feasible the derivatization at the C2′ position of (−)-quinine.

This synthesis requires a total of five separate one-pot operations and five purifications by column chromatography, plus two additional operations for the conversion of the undesired **epi-20** to the desired **20**. The total yield of (−)-quinine was 14%.

## Methods
### General information
All reactions were carried out under argon atmosphere and monitored by thin-layer chromatography using Merck 60 F254 precoated silica gel plates (0.25 mm thickness). Specific optical rotations were measured using a JASCO P-2200 polarimeter. FT-IR spectra were recorded on a JASCO FT/IR-4600HC1 spectrometer. ¹H and ¹³C NMR spectra were recorded on an Agilent-400 MR (400 MHz for ¹H NMR, 100 M Hz for ¹³C NMR) instrument. Data for ¹H NMR are reported as chemical shift (δ ppm), integration multiplicity (s = singlet, d = doublet, t = triplet, q = quartet, septet = sep, dd = doublet of doublets, ddd = doublet of doublet of doublets, ddt = doublet of doublet of triplets, dt = doublet of triplets, dq = doublet of quartets, m = multiplet, brs = broad singlet, brd = broad doublet, brt = broad triplet), coupling constant (Hz), Data for ¹³C NMR are reported as chemical shift. High resolution ESI-TOF mass spectra were measured by Themo Orbi-trap LTQ XL instrument. HPLC analysis was performed on a HITACHI Elite LaChrom Series HPLC, UV detection monitored at appropriate wavelength respectively, using CHIRALPACK® IB (0.46 cm × 25 cm). Melting point was measured using Yanaco MP-J3. Flash chromatography was performed using silica gel 60 N (spherical) or silica gel 60 N (spherical) NH₂ of Kanto Chemical Co. Int., Tokyo, Japan. All reagents were purchased from commercial sources (Aldrich, FUJIFILM Wako chemicals, Kanto Chemical, TCI).

## Data availability
Experimental procedures, characterizations, spectra are available in the Supplementary Information.

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

## Acknowledgements

This work was supported by JSPS KAKENHI Grant Number JP19H05630 (Y.H.).

## Author contributions

T.T. performed the experiments. Y.H. conceived the concept and prepared the manuscript with feedback from T.T.

## Competing interests

The authors declare no competing interests.
