## [Peer Review File · Nature Communications]

REVIEWER COMMENTS

Reviewer #1 (Remarks to the Author):

In this manuscript, Hayashi and coworkers have described a five-pot synthesis of (-)-quinine protocol mediated by organocatalyst. Using one-pot operations, the authors afford an efficient and direct access to (-)-quinine, Br(C2')-(-)-quinine and its relative derivatives. A sequential diphenylprolinol silyl ether-mediated asymmetric Michael reaction, aza-Henry reaction and elimination in first lead to the chiral piperidine skeleton. In the fourth pot, 2,4-dibromo-6-methoxy quinoline is used to introduce quinoline moiety. This protocol could efficiently yield (-)-quinine with 17% total yield using five pot operations. Thus, I recommend this manuscript could be accepted by Nature Communication after minor revision only if the following points are addressed.

- 1) In page 6, paragraph 3, line 6: "In order to synthesize C2' derivatives of quinine" C2' might be C4'.
- 2) In the second pot, considering the deprotection and reprotection included in this procedure, can 11 be directly reduced to 15 using hydrogenation condition or other conditions?
- 3) How much can this five-pot protocol scale up, and how efficient this protocol is as compared to previous reports? These could be included in the manuscript.

Reviewer #2 (Remarks to the Author):

This manuscript reports a new total synthesis of the well-known alkaloid quinine. The authors claimed a pot-economy approach to this natural product. Key to the synthesis is an organocatalyzed asymmetric Michael/aza-Henry/hemiaminalization domino reaction which was developed by the same group in 2005 (ref. 8). This crucial transformation was applied to well-designed substrates to efficiently afford an advanced intermediate 10. Another feature of the synthesis includes the use of di-bromo 10 in the fragment coupling with aldehyde 18, which not only suppressed the C2-addition by-product, but also allowed for analog preparation (as shown in Scheme 3). Collectively, this study provides an interesting entry to cinchona alkaloids in an efficient manner.

While the chemistry is well illustrated in the present manuscript, one major concern of the referee is the scalability of the whole synthetic route. Because of the described efficiency and convenience (5

pots and 17% overall yield) of the total synthesis, it would not be difficult for the authors to conduct each pot of the synthesis on large gram scales and to obtain at least gram quantity of the target quinine. However, only 1.7 mg of the final natural product was synthesized according to the supporting information. In this regard, the result seems inconsistent with the conclusions. Thus, the authors are encouraged to examine the scalability of the pot-economic synthesis and supplement the corresponding results in the main text (in particular in Scheme 2) as well as in the SI. This manuscript is recommended for publication, provided the above issue is addressed.

Reviewer #3 (Remarks to the Author):

Cinchona alkaloid (-)-quinine is an important molecule widely used in both drug discovery and catalyst development. Although the asymmetric synthesis of (R)-quinine have been accomplished by several research groups before, Hayashi and Terunuma described an efficient and very much different synthetic strategy in this manuscript. They have finished the enantioselective total synthesis of (R)-quinine though only five pot reaction pots. Two to five steps of reactions were performed in each pot, greatly simplifying the reaction procedure and minimizing the generation of chemical waste. The overall yield is 17%, which is an excellent result. I think this is an attractive work that showcase the powerfulness of one-pot reaction strategy applied in total synthesis. The Br-containing quinine could also be obtained though one-pot reaction of compound 20, which show great synthetic potential in quinine modification at C2 position. The synthetic route of total synthesis is rationally designed and the manuscript is well-written. Based on the above reasons, I believe this manuscript is suitable for publication in Nature Communication after the minor revisions listed below:

1. The pot Number 2: I notice that two similar conditions were used for the sequential reduction of hemiaminal and enamine on compound 11. Could the two moieties be reduced under a single conduction (such as TFA, NaBH(OAc)₃), so that the reaction procedures could be further simplified?
2. The pot Number 4: The diastereoselectivity in LDBBA reduction step should be presented (ratio of 20/epi-20).
- 3, There are some impurities in the NMR spectrum for the synthesized intermediates. Please purify them and update the NMR data.

Reviewer 1: Thus, I recommend this manuscript could be accepted by Nature Communication after minor revision only if the following points are addressed.

Answer

Thank you for the excellent comment for the acceptance after the minor revision.

Comment 1. In page 6, paragraph 3, line 6: "In order to synthesize C2' derivatives of quinine" C2' might be C4'.

Answer

I think C2' is correct, and I did not modify the text.

Comment 2. In the second pot, considering the deprotection and reprotection included in this procedure, can **11** be directly reduced to **15** using hydrogenation condition or other conditions?

Answer

*Thank you for the nice comment. We examined the hydrogenation and other conditions to reduce **11**, but we did not have good results. Hydrogenation, for instance, afforded the low diastereoselectivity. That is why we have to use de-protection and reprotection procedure. We added the following sentence in page 4.*

*In this transformation, procedures of deprotection and reprotection of Boc group were involved. The stereoselective reduction of Boc protected alkene **11** did not proceed well, while stereoselective reduction the iminium ion **14** was successful. Thus, the deprotection step of Boc group is necessary.*

Comment 3. How much can this five-pot protocol scale up, and how efficient this protocol is as compared to previous reports? These could be included in the manuscript.

Answer

Thank you for the comment. We investigated the reaction larger scale. Gram scale reaction also proceeded in a similar yield. We revised the yield based on the larger scale. In Scheme 2, the yield and scale were described to indicate the scalability of the reaction. Total yield in the gram scale synthesis is 14%, and we changed the yield from 17% to 14% in the Scheme 2 and the text. We also described the gram scale synthesis in SI.

Reviewer 2:

Comment.

While the chemistry is well illustrated in the present manuscript, one major concern of the referee is the scalability of the whole synthetic route. Because of the described efficiency and convenience (5 pots and 17% overall yield) of the total synthesis, it would not be difficult for the authors to conduct each pot of the synthesis on large gram scales and to obtain at least gram quantity of the target quinine. However, only 1.7 mg of the final natural product was synthesized according to the supporting information. In this regard, the result seems inconsistent with the conclusions. Thus, the authors are encouraged to examine the scalability of the pot-economic synthesis and supplement the corresponding results in the main text (in particular in Scheme 2) as well as in the SI. This manuscript is recommended for publication, provided the above issue is addressed.

Answer: Thank you for the excellent comment. This is a same comment as comment 3 of the reviewer 1. Please see the answer to the reviewer 1.

Reviewer 3

General Comment

Based on the above reasons, I believe this manuscript is suitable for publication in Nature Communication after the minor revisions listed below:

Answer

Thank you for the excellent comment for the acceptance after the minor revision.

Comment 1

The pot Number 2: I notice that two similar conditions were used for the sequential reduction of hemiaminal and enamine on compound 11. Could the two moieties be reduced under a single conduction (such as TFA, NaBH(OAc)₃), so that the reaction procedures could be further simplified?

Answer

Thank you for the comment. It is a good point. We also thought the same things. We examined the conditions suggested by this reviewer, but the reaction did not proceed well. We wrote the fooling sentence in page 4.

*We employed two reducing reagents such as Et_3SiH and $\text{NaBH}(\text{OAc})_3$ for the reduction of hemiacetal (**10**→**11**) and iminium ion (**13**→**14**). It would be synthetically simple to use a single reducing reagent in a one-pot reaction. Although Et_3SiH can act as a reducing reagent in both reactions, the second reaction was slow in the low yield with the partial deprotection of TBDPS group. In the case of $\text{NaBH}(\text{OAc})_3$ the reaction from **10** to **11** did not proceed well with the generation of several byproducts. On the other hand, the sequential use of two reducing reagents gave a good result in the second pot reaction.*

Comment 2

The pot Number 4: The diastereoselectivity in LDBBA reduction step should be presented (ratio of 20/epi-20).

Answer

Thank you for the comment. We wrote the yield of 20 and epi-21 in the reduction of LDBBA in the Scheme 2. The text was modified as follows:

*Although the undesired diastereomer **epi-20** was obtained, **epi-20** can be converted to the desired isomer **20** via oxidation with DMP (90%) and stereoselective reduction using LDBBA (lithium diisobutyl-tert-butoxyaluminum hydride)^{52,53}, in which **20** was obtained in 85% with **epi-20** (14%).*

Comment 3

There are some impurities in the NMR spectrum for the synthesized intermediates. Please purify them and update the NMR data.

Answer

We are sorry that our sample contains impurities. We updated the NMR data in SI.

REVIEWERS' COMMENTS

Reviewer #1 (Remarks to the Author):

In the revised manuscript, all the points of three reviewers have been addressed. Gram scale reaction proceeded well in a similar yield, which has been added in the manuscript and SI. Other questions for the pot 2 and 5 are also addressed well, and these points have also been added in the manuscript. Thus, I recommend this manuscript could be accepted by Nature Communication for publication

Reviewer #2 (Remarks to the Author):

The authors have addressed my concerns in this revised version, which is thus recommended for publication.

Reviewer #3 (Remarks to the Author):

The authors have fully addressed the concerns of this reviewer. The publication of this work in Nature Communications is therefore recommended.